# The Role of Activation of PI3K/AKT/mTOR and RAF/MEK/ERK Pathways in Aggressive Pituitary Adenomas—New Potential Therapeutic Approach—A Systematic Review

**DOI:** 10.3390/ijms241310952

**Published:** 2023-06-30

**Authors:** Aleksandra Derwich, Monika Sykutera, Barbara Bromińska, Błażej Rubiś, Marek Ruchała, Nadia Sawicka-Gutaj

**Affiliations:** 1Department of Endocrinology, Metabolism and Internal Medicine, Poznan University of Medical Sciences, 60-355 Poznan, Poland; aleksandra.derwich@student.ump.edu.pl (A.D.);; 2Doctoral School, Poznan University of Medical Sciences, 60-812 Poznan, Poland; 3Department of Clinical Chemistry and Molecular Diagnostics, Poznan University of Medical Sciences, 60-355 Poznan, Poland

**Keywords:** pituitary tumor, aggressive pituitary adenoma, RAF/MEK/ERK pathway, PI3K/AKT/mTOR pathway

## Abstract

Pituitary tumors (PT) are mostly benign, although occasionally they demonstrate aggressive behavior, invasion of surrounding tissues, rapid growth, resistance to conventional treatments, and multiple recurrences. The pathogenesis of PT is still not fully understood, and the factors responsible for its invasiveness, aggressiveness, and potential for metastasis are unknown. RAF/MEK/ERK and mTOR signaling are significant pathways in the regulation of cell growth, proliferation, and survival, its importance in tumorigenesis has been highlighted. The aim of our review is to determine the role of the activation of PI3K/AKT/mTOR and RAF/MEK/ERK pathways in the pathogenesis of pituitary tumors. Additionally, we evaluate their potential in a new therapeutic approach to provide alternative therapies and improved outcomes for patients with aggressive pituitary tumors that do not respond to standard treatment. We perform a systematic literature search using the PubMed, Embase, and Scopus databases (search date was 2012–2023). Out of the 529 screened studies, 13 met the inclusion criteria, 7 related to the PI3K/AKT/mTOR pathway, and 7 to the RAF/MEK/ERK pathway (one study was used in both analyses). Understanding the specific factors involved in PT tumorigenesis provides opportunities for targeted therapies. We also review the possible new targeted therapies and the use of mTOR inhibitors and TKI in PT management. Although the RAF/MEK/ERK and PI3K/AKT/mTOR pathways play a pivotal role in the complex signaling network along with many interactions, further research is urgently needed to clarify the exact functions and the underlying mechanisms of these signaling pathways in the pathogenesis of pituitary adenomas and their role in its invasiveness and aggressive clinical outcome.

## 1. Introduction

Pituitary neuroendocrine tumors (PitNETs), or pituitary adenomas (PAs), are a heterogeneous group of central nervous system lesions that are mostly benign [1]. They are the second most common intracranial neoplasms, with an estimated incidence of 16.7% in the general population [2], (~78 to 116 cases per 100,000 people) [3]. According to autopsy studies, the prevalence is even higher and observed in 20–25% of cases [4,5].

The most common pituitary tumors are prolactinomas (40–66%), followed by clinically non-functioning pituitary adenomas (NFPAs, 15–43%), somatotropinomas (acromegaly or gigantism, 8–16%), corticotropinomas (Cushing’s disease, 2–6%), and rarely thyrotropinomas (<1%) or gonadotropinomas [6]. Functioning pituitary tumors (hormone-producing) are usually diagnosed earlier than non-functioning ones [7]. The data obtained from pathological examination show that only 14% of pituitary adenomas are invasive, and 2% are aggressive [8]. Metastatic PitNETs are rare (0.1–0.5% of cases). They are classified according to their size in microtumors (<1 cm), macrotumors (≥1 cm), or giant tumors (≥4 cm). The clinical outcome of pituitary tumors differs greatly. Some remain still for a long time, many grow slowly, and in rare cases, rapid tumor growth may also be observed [9]. First-line treatment is surgical intervention, except from prolactinomas, which usually respond well to pharmacotherapy with tumor shrinkage and decrease in secreted hormone levels. The aim of surgical intervention is to remove the tumor mass as well as eliminating the effect of the tumor compression on the surrounding tissues. Patients with prolactinoma who do not respond to pharmacological treatment or experience severe side effects should also undergo surgical resection. Almost 90% of pituitary tumors can be safely excised using a transsphenoidal approach under fluoroscopic guidance and microsurgical techniques. However, even in approximately 20 to 30% of patients, the surgery is insufficient, and additional treatment is necessary such as subsequent surgeries, radiotherapy, and pharmacotherapy [3]. Radiotherapy is usually used in the case of residues or recurrences in inoperable sites. The risk of tumor progression in the presence of a residual tumor is increased [3]. Aggressive behavior can show up more than ten years after the diagnosis of a pituitary tumor, and so far, there are no markers able to predict this [10]. Some of the histological variants can be associated with more clinical aggressiveness. High-risk lesions include sparsely granulated somatotroph adenomas, silent corticotroph adenomas and Crooke’s cell adenoma [8].

Invasion is defined as the tumor invading surrounding structures such as the cavernous sinuses and sphenoid sinus as well as the focal or extensive bones, which accounts for 22–55% of PitNET/PA cases [3]. The frequency of aggressive tumors is hard to assess [11]. They often show one of three markers: Ki-67 ≥ 3%, and/or increased mitosis, and/or expression of p53 [12]. What is more, invasiveness alone is not synonymous with pituitary tumor aggressiveness, although it is a significant determinant of incomplete tumor resection. Aggressive pituitary tumors are almost always macroadenomas at clinical presentation. However, pituitary tumor size at presentation does not equate to the potential for aggressive behavior [3]. Aggressive pituitary tumors usually respond poorly to conventional treatment for non-aggressive tumors [11]. Current approaches in managing aggressive pituitary tumors (APTs) are described in the treatment section. The 5-year survival rate for pituitary tumors is relatively high and in general amounts to 97%, but in the group of patients with metastatic PitNETs is estimated to be just over 81%. Acromegaly is associated with a twofold increase in mortality, but only when the excessive GH secretion is uncontrolled by the treatment. Cushing’s disease is associated with up to a fourfold increase in mortality. Effective treatment reduces mortality, but mortality nonetheless remains elevated in comparison to the general population [8].

The current PitNET classification is based on the pituitary cell lineages and immunohistochemical staining of the hormone content and primary transcription factors in the tumor [13]. Pituitary cell lineages are determined by the expression of the hormones of the anterior pituitary and pituitary-specific transcription factors [14]. Steroidogenic factor 1 (SF1), encoded by the NR5A1 gene, regulates the differentiation of gonadotrophs producing follicle-stimulating hormone (FSH) and luteinizing hormone (LH). Pituitary transcription factor 1 (PIT1), encoded by the gene POU1F1 determines the development of cells that produce growth hormone (GH), prolactin (PRL), and thyroid-stimulating hormone (TSH). The transcription factor T-box TPIT, encoded by the TBX19 gene, is responsible for the development of corticotrophs producing adrenocorticotropic hormone (ACTH), encoded by the POMC gene [15,16]. Null cell adenomas do not secrete hormones and are negative for the expression of lineage transcription factors [17].

Despite numerous studies and advances in predictive classification, no pathological marker has been identified so far that could reliably predict the behavior of a pituitary tumor [18]. There is evidence at the molecular level that pituitary tumors accumulate abnormalities in molecular pathways over time that contribute to their progression from benign adenomas to aggressive recurrent pituitary tumors and, in exceptional cases, to pituitary cancer [19,20,21]. Many attempts have been made to explore the potential mechanisms involved in PitNET invasion. It is known that signaling pathways are crucial for cell homeostasis, and its dysregulation dictates cell-cycle arrest or malignant transformation. The roles of RAF/MEK/ERK and PI3K/AKT/mTOR pathways in pituitary tumorigenesis are still studied and, due to their complexity, far from being fully understood. Since current treatment options for aggressive pituitary tumors are insufficient to control tumor growth and often confer significant morbidity, therapeutic targeting of molecular pathways may offer alternative treatments for these patients. In this review, we present data on studies on the role of these pathways in pituitary tumorgenesis and discuss interesting therapeutic approaches embracing PI3K/AKT/mTOR pathway and Raf/Mek/ERK pathway in managing APTs.

### 1.1. Current Knowledge of Pituitary Adenoma Pathogenesis

The exact pathogenesis of pituitary adenomas is not well understood but is thought to be related to a combination of genetic and environmental factors [17]. These tumors are most likely monoclonal and arise as a result of the uncontrolled expansion of a single, somatically mutated cell [22,23,24,25]. Somatic mutations occur in specific genes, such as GNAS, USP8, and ATRX [24,26,27,28,29,30]. For example, mutations in GNAS have been found in a subset of GH-secreting adenomas, while mutations in USP8 have been associated with Cushing’s disease [29]. Other somatic changes suggested as being associated with pituitary tumors include PIK3CA amplification, IDH1 mutations, TP53 gene mutation in pituitary carcinomas and corticotropinomas, and HMGA2 amplification in prolactinomas [29,31,32,33,34]. However, evidence of polyclonalism in some pituitary adenomas can also be found in the literature [22,35]. The exact cell origin is unknown and may vary between tumors, but PAs are usually well differentiated histologically [24]. Candidates for a cell of origin are differentiated or progenitor anterior pituitary cells or pituitary stem cells [24,36,37].

Epigenetic modifications have become increasingly important in understanding tumorigenesis, as most pituitary tumors are sporadic with no known genetic driver. Epigenetic silencing of several tumor suppressor genes has been documented in PitNETs [38]. The pituitary epigenetic changes at the chromatin (pretranscription) and RNA levels (post-transcription) are especially crucial in determining clinical characteristics such as subtype differentiation and local invasion. Methylation of CpG islands in CDKN2A is seen in up to 90% of sporadic pituitary tumors with loss of expression of p16 observed in immunohistochemistry. Methylation of CpG islands in the RB1 promoter in sporadic tumors is significantly associated with loss of expression observed in immunohistochemistry [24,29,36,39,40]. Histone acetyltransferases p300 have been found to upregulate the human pituitary tumor transforming gene (PTTG1) [41]. Several miRNAs have been identified as tumor suppressors in pituitary tumors [42]. miR-375 is highly expressed in prolactin-secreting pituitary adenomas (prolactinomas) and has been shown to regulate prolactin secretion by targeting the prolactin gene. Other miRNAs, such as miR-26b and miR-142-3p, are associated with GHPA- and ACTH-secreting PA. Higher levels of miR-16-5p, miR-145-5p, and miR-7g-5p have been found in the plasma of patients with Cushing’s disease (CD) [43] A study comparing patients with invasive and non-invasive PitNET showed that miR-200a expression was increased in invasive samples [44] In profiling CpG island methylation status that included the genes encoding RB1, p14(ARF), p15(INK4b), p16(INK4a), p21(Waf1/Cip1), and p27(Kip1), 88% (30 of 34) of the adenomas displayed methylation of at least one of such cell-cycle regulatory genes [40].

Another study analyzing bioinformatic data revealed the significant signaling pathways and networks associated with pituitary adenomas [45]. Signaling pathways and networks that were found to be significantly associated with a pituitary adenoma included mitochondria dysfunction, oxidative stress, cell-cycle dysregulation, and the MAPK-signaling pathway. Uncontrolled progression through the cell cycle is a primary driver of tumorigenesis. Progression through the cell cycle depends mainly on fluctuations in the concentration of cyclins (CDKs) and their inhibitors (CDKI) achieved by programmed degradation of these proteins by proteolysis within the ubiquitin-proteasome system. There is also a transcriptional regulation of cyclin expression, probably dependent on CDK phosphorylation. Then, the level of tumor suppressors involved in cell-cycle checkpoints, such as p16 and Rb, is reduced, while the level of proteins promoting cell-cycle progression, such as cyclin, is increased [46]. Deregulation in the Rb/p16/cyclin D1/CDK4 pathway is present in up to 80% of pituitary adenomas [47].

There are also new data on the proteomics of PitNETs. Two large studies [48,49], including cohorts of 134 and 200 patients, respectively, revealed novel information on PitNETs’ proteogenomic characterization. Proteomic-based classification of PitNETs identified seven clusters, among which tumors overexpressing epithelial-mesenchymal transition (EMT) markers clustered into a more invasive subgroup [49]. Novel tumor-related genes, such as *AMIGO2*, *ZFP36*, *BTG1*, and *DLG5*, were identified, which may lead to further investigations into identifying novel therapeutic targets.

### 1.2. The Challenge

Despite numerous studies on prognostic parameters related to PA subtypes and metastasis, to date no signaling molecules have been reported as markers. Thus, early identification of the increased potential for aggressiveness and invasiveness is still challenging. 

According to the most recent European Endocrine Society guidelines on an APT, the first choice of medical therapy after tumor growth is temozolomide (TZM). However, its efficacy is not satisfactory. Tumor volume shrinkage was reported in 37% of cases [12]; meanwhile, in a recently reported survey, a complete response was demonstrated in 9.6% of cases, with partial response in 30.1% and stable disease (SD) in 28.1% [50]. Reduced effectiveness of TZM is usually explained by high O(6)-methylguanine methyltransferase (MGMT) content in tumor tissue. MGMT is an enzyme which can counteract the cytotoxic effect of TZM [51]. Other more common options are radiotherapy (RT) [results after first RT: complete response (CR)-3.2%; partial response (PR)-41.9%; SD-47.6%], RT combined with TZM, somatostatin analogues, peptide receptor radionuclide therapy (PRRT), bevacizumab, immune checkpoint inhibitors (ICI), other cytostatic drugs, mTOR inhibitor, and tyrosine kinase inhibitors. For all those methods, treatment efficacy is either insufficient or difficult to ascertain [50]. Due to the rarity of APT and the nonhomogeneous structure of research concerning novel therapies, it is hard to draw solid conclusions in favor of or against any drug. Moreover, most studies concerning new treatment options reported very small groups of patients, so usefulness of the methods is hard to estimate [50]. Other currently used drugs, along with interesting therapeutic approaches and challenges embracing PI3K/AKT/mTOR pathway and Raf/Mek/ERK pathway in APT, are described in the treatment section.

Additionally, animal models currently used in pituitary tumor studies display several limitations. There are four main types of animal models. The first one is the cell line-based xenograft (CDX); the second is the patient-derived xenograft (PDX); then, the environmentally induced model; and, finally, the genetically engineered mouse (GEM) model. In vitro cultured animal-derived cell lines and CDX are most commonly applied in pituitary research but demonstrate some disadvantages. Firstly, the tissue architecture and tumor microenvironment are different compared to human tumors. Secondly, CDX shows loss of genetic heterogeneity because of the long-term in vitro propagation. When it comes to PDX models, they are rarely used in the case of PA. There are no commercial human PA cell lines; therefore, primary human PA cell cultures are relatively hard to obtain and can only be used for short-term studies [52]. Moreover, most cell lines in PA research are rat- or mouse -derived. The pathophysiology of a rodent’s pituitary gland is different than in humans [14]. There are promising new approaches aimed at establishing reliable human PA models, such as pituitary induction methods from human-induced pluripotent stem cells [53].

To summarize, there are three vital obstacles to overcome concerning APT: lack of reliable prognostic markers, insufficient efficacy of medical therapies, and difficulties with pituitary tumor models.

### 1.3. Role of PI3K/AKT/mTOR Pathway in Cancerogenesis

Understanding the mechanisms of receiving and integrating extracellular signals in cells and triggering a cascade of intracellular signals that affect cell growth and metabolism are essential for developing effective diagnostic and treatment strategies. The PI3K/AKT/mTOR pathway links growth factors, nutrients, and energy available to cell survival, growth, motility, and proliferation [54]. The biological role of the PI3K/AKT/mTOR signaling pathway in the development of cancer is shown in Figure 1. Phosphoinositide 3-kinases (PI3Ks), also called phosphatidylinositol 3-kinases, are a family of related intracellular signal transducer enzymes capable of phosphorylating the 3-position hydroxyl group of the inositol ring of phosphatidylinositol (PtdIns). The pathway, with oncogene PIK3CA and tumor suppressor gene PTEN, is implicated in the sensitivity of cancer tumors to insulin and IGF-1. Human cells express three classes of PI3K enzymes. There are four class I catalytic isoforms (p110α, β, γ, and δ encoded by PIK3CA, PIK3CB, PIK3CG, and PIK3CD, respectively), three class II PI3Ks (PI3K-C2α, β, γ encoded by PIK3C2A, PIK3C2B and PIK3C2G, respectively), and a single class III PI3K (hVPS34, encoded by PIK3C3) [55]. AKT, also known as protein kinase B (PKB), is a serine-threonine protein kinase identified as one of three isoforms, AKT1, AKT2, and AKT3, which are encoded by three distinct genes PKBα, PKBβ, and PKBγ [56], respectively. AKT1 is involved in cellular survival pathways by inhibiting apoptotic processes. It is capable of inducing protein synthesis pathways and is a key signaling protein in the cellular pathways that lead to skeletal muscle hypertrophy and general tissue growth. AKT2 is an important signaling molecule in the insulin signaling pathway. It is required to induce glucose transport. The role of AKT3 is less clear, though it appears to be predominantly expressed in the brain [56,57,58].

mTOR (mammalian target of rapamycin) is a 289 kDa serine/threonine kinase, a member of the phosphatidylinositol 3-kinase-related kinase family of protein kinases. It is encoded by the MTOR gene [59,60]. PI3K activation phosphorylates and activates AKT, localizing it in the plasma membrane. AKT can affect a number of downstream effects such as activating CREB, inhibiting p27, localizing FOXO in the cytoplasm, activating PtdIns-3ps, and activating mTOR which can affect transcription of p70 or 4EBP1 [61,62,63]. The mTOR kinase is crucial for maintaining cellular homeostasis by integrating multiple cell-signaling pathways. mTOR acts as a sensor of cellular energy compounds, ATP levels, and redox status [64]. Dysregulation of the mTOR kinase pathway is an essential factor in the pathogenesis of various human diseases, especially cancer [62], where it controls cell metabolism by altering key metabolic enzymes’ expression and/or activity. Abnormal activation of the mTOR pathway through stimulation of oncogenes or loss of tumor suppressors contributes to tumor growth, angiogenesis, and metastasis. In several human cancers, the mutations in the mTOR gene that confer constitutive activation of mTOR signaling have been identified [54,62,65,66,67].

mTOR comprises two functionally distinct protein complexes: mTOR complex 1 (MTORC1) and mTOR complex 2 (MTORC2). MTORC1 primarily regulates cell growth, metabolism, and protein synthesis, while MTORC2 is involved in regulating cell survival, cytoskeletal dynamics, and metabolism [66]. The mTORC1 consists of mTOR, mLST8 (mammalian lethal with SEC13 protein 8/G protein β-subunit-like protein GβL), RAPTOR (regulatory-associated protein of mTOR), and two non-core components: PRAS40 (proline-rich AKT1 substrate 1) and DEPTOR (DEP domain-containing mTOR-interacting protein) [54,68,69,70,71]. The mTORC2 includes mTOR, Rictor (rapamycin-insensitive companion of mTOR), mLST8, mSin1 (mammalian stress-activated protein kinase-interacting protein 1), Protor (protein observed with Rictor/PRR5, proline-rich protein 5), and DEPTOR [62].

mTORC1 is activated by the PI3K/AKT pathway and inhibited by the TSC1/TSC2 complex [62]. Activation of MTORC1 is dependent on the presence of sufficient nutrients, such as amino acids (leucine, arginine), and growth factors (insulin, insulin-like growth factor 1 (IGF-1)), as well as cellular energy status [65]. mTORC1 is translocated from the cytoplasm to the lysosomal surface in response to nutrients and activated by growth factors through PI3K-AKT signaling. Growth factors, such as insulin, activate AKT4 through the cognate receptor, phosphoinositide-dependent kinase 1 (PDK1), and PI3K. AKT inhibits the TSC1-TSC25 complex, which is a GTP-activating protein (GAP) for the small GTPase RHEB6. GTP-bound RHEB directly binds and activates mTORC1 in the lysosome [65]. Low levels of energy compounds in the cell, low levels of growth factors, the low redox potential of the cell, caffeine, rapamycin, farnesylthiosalicylic acid (FTS), and curcumin [72,73] contribute to mTORC1 inhibition. p70-S6 kinase (S6K1) and eukaryotic translation initiation factor 4E binding protein 1 (eIF4E binding protein 1, 4E-BP1) are the best-characterized substrates of the mTORC1 complex [64]. When activated, mTORC1 promotes protein synthesis by phosphorylating downstream targets such as S6K1 and 4EBP1, which regulate translation initiation and protein synthesis.

S6K1 can also participate in a positive feedback loop by phosphorylating the mTOR kinase molecule. The eIF4E binding protein is inhibited by mTORC1 to enhance translation, including the translation of metabolic enzymes and metabolism-related transcription factors. Additionally, mTORC1 and S6K are able to directly regulate metabolic enzymes [54,62,65,66,74,75].

mTORC2 is an essential regulator of cell cytoskeletal function. It interacts with the proteins F-actin, paxillin, RhoA, Rac1, Cdc42, and protein kinase Cα (PKCα) [76] and promotes metabolism mainly through the activation of AKT kinase. S6K and AKT regulate metabolic enzymes and activate key metabolic transcription factors such as MYC, α hypoxia-induced factor 1 (HIF1 α) and α HIF2, transcription factors FOXO, and the regulatory element sterol 1 binding protein (SREBP1). Activation of mTORC2 is less well understood compared to mTORC1, but it is known to be regulated by growth factors, such as insulin and IGF-1, and phosphoinositide 3-kinase (PI3K) signaling [77]. It phosphorylates PKC-α, AKT, and paxillin and regulates the activity of the small GTPases Rac and Rho-associated with cell survival, migration, and regulation of the actin cytoskeleton [54,76]. Growth-factor signaling alone is sufficient to activate mTORC2, but its mechanism is still incompletely understood [78].

mTOR plays a significant role in physiology and pathology. Its role has been identified in the regulation of tissue regeneration, regulatory T cell differentiation and function, and diverse types of cancer, including hematologic malignancies, prostate, breast, skin, and head and neck cancers [62,79,80,81,82,83,84,85]. AKT is amplified in a subset of human cancers, such as breast and ovarian cancers [54]. 4EBP1 expression has been found to be associated with poor prognosis in breast, colon, ovarian, and prostate cancers, and the phosphorylation of 4EBP1 seems to be associated with chemoresistance in ovarian cancer [86]. High *PI3K* activity has been implicated in cell transformation and tumor progression in ovarian, gastrointestinal, breast, and prostate cancers [87]. mTOR signaling is activated in anaplastic and medullary thyroid cancer [88,89,90]. In a meta-analysis of the genetic polymorphisms of mTOR and cancer risk [91], mTOR rs2295080 G allele was associated with a significantly higher risk of acute leukemia in the recessive model and a lower risk of genitourinary cancers in the dominant model. The potential of mTOR has been discussed in chronic myeloid leukemia (CML) and acute myeloid leukemia (AML) since it is known that activation of the mTOR pathway is associated with deregulated production of malignant lymphoid cells and chemotherapeutic resistance in acute lymphoblastic leukemia (ALL). Treatment with dual PI3K/mTOR inhibitors or mTOR kinase inhibitors alone or in combination with conventional ALL therapies or with targeted drugs for different cellular cascades inhibits cell proliferation and induces apoptosis by blocking distinct mechanisms of cell survival in ALL [79,83,92]. Genetic alterations or ultraviolet (UV) exposure results in the dysregulation of the PI3K/AKT/mTOR pathway in melanocytes, basal cells, squamous cells, or Merkel cells, which leads to the development of melanoma, basal cell carcinoma, cutaneous squamous cell carcinoma, or Merkel cell carcinoma [93]. PI3K/AKT/mTOR signaling is active in over 90% of squamous cell carcinomas of the head and neck [80]. Clinical studies already evaluate that using PI3K/AKT/mTOR pathway inhibitors in breast, skin, and prostate cancer is a promising strategy to fight against these diseases. [81,82,85,94].

### 1.4. Role of Raf/MEK/ERK in Cancerogenesis

The Raf/MEK/ERK signaling pathway (also known as the MAPK/ERK pathway) is crucial in regulating cell growth and proliferation. It is essential in cellular differentiation, survival, and apoptosis. Dysregulation of this pathway has been implicated in various human diseases, including cancer, neurodegenerative disorders, and cardiovascular diseases [95,96]. The biological role of the Raf/MEK/ERK signaling pathway in the development of cancer is shown in Figure 1. The Raf/MEK/ERK pathway consists of three central protein kinases: RAF, MEK, and ERK. ERKs (extracellular-signal-regulated kinases) promote cell proliferation, survival, and metastasis, mainly through activation by epidermal growth factor receptor (EGFR) and Ras small guanosine triphosphatases (GTPases) [97]. ERK has two homologs, ERK1 and ERK2, which are serine-threonine kinases, and their activity is positively regulated by phosphorylation and mediated by MEK1 and MEK2 [98]. In the human genome, ERK1 and ERK2 are encoded by the MAPK3 and MAPK1 genes, respectively. Phosphorylated ERK (pERK) is a critical component of the downstream signaling pathway Ras/Raf/MEK/ERK [99]. After phosphorylation, it is translocated to the nucleus, leads to gene expression changes, and regulates various transcription factors [97,99]. The discovery of the Raf/MEK/ERK signaling pathway is associated with the identification of the mitogen-activated protein kinase (MAPK). ERK activation requires phosphorylation on threonine and tyrosine residues and is inactivated by phosphorylation by MAP kinase (MKK) or MAP/ERK kinase (MEK) [96]. RAF is a direct regulator of the MKK and is able to restore the activity of protein phosphatase 2A-inactivated MKK [96,100]. MEK1 and MEK2 are encoded by the MAP2K1 and MAP2K2 genes, respectively. Similar to ERK1 and ERK2, MEK proteins exhibit similar functions driving ERK activation. The MKKK family is the largest, with more than 20 genes identified to date, which include RAF kinases [101]. They are encoded by three isoforms: *ARAF*, *BRAF*, and *CRAF* [96]. RAF kinases exhibit more divergent physiological functions, unlike ERK and MEK. RAF kinases are protomers that combine into functional homo- or heterodimers. The small RAS GTPase is usually considered part of the Raf/MEK/ERK signaling module. The human genome encodes three RAS genes—*HRAS*, *KRAS*, and *NRAS*, whose expression and function vary in different tissues and at different stages of development [96,97,98,99].

Signal transduction via the Raf/MEK/ERK pathway is initiated by binding various ligands to the tyrosine kinase receptor, specifically growth factor receptors such as EGFR. The receptor–ligand interaction promotes tyrosine kinase receptor dimerization, activation, and autophosphorylation on several tyrosine residues in the intracellular domain. Phosphorylated tyrosine residues are recognized by proteins containing the SH2 or PTB domain, such as Grb2. Grb2, in turn, recruits the GEF SOS protein, which facilitates the exchange of GDP to GTP on RAS proteins [96,102]. The GTP-loaded RAS marks the start of a three-stage phosphorylation cascade through RAF recruitment and activation, which is activated in a process involving plasma membrane recruitment, dimerization, and subsequent phosphorylation. RAF phosphorylates and activates the MEK1/2 protein kinases. MEK1/2 phosphorylates ERK1 on T202/Y204 and ERK2 on T183/Y185, respectively. Phosphorylated ERK1 and ERK2 are then translocated into the nucleus, activating transcription factors that regulate various physiological processes by altering the gene expression profile [63,96,98,100,103,104]. Raf/MEK/ERK pathway dysregulation is associated with cancer oncogenesis, as evidenced by the facilitation of tumor proliferation, survival, invasion, metastasis, extracellular matrix degradation, and angiogenesis. Mutations in the components of the Raf/MEK/ERK signaling pathway have been linked to different types of cancer. Most cancers develop with aberrant activation of the RAF/MEK/ERK pathway either due to oncogenic mutations in its components or its upstream regulators such as RAS and EGFR. BRAF is often mutated in melanoma, papillary thyroid, colorectal, and ovarian cancers. *BRAF* (predominantly *BRAF*^V600E^) is the most common molecular alterations in papillary thyroid carcinoma. The central role in its oncogenesis is constitutive activation of the MAPK signaling pathway through mutations or fusion of its essential proteins such as receptor tyrosine kinases (RTKs, which includes RET, ALK, VEGFR, and TRK), RAS, RAF, MEK, and ERK [105]. MEK genes are mutated in less than 1% of all cancers, while mutations in ERK genes are sporadic. Abnormal activation of the Raf/MEK/ERK signaling pathway may also be caused by mutations in RAS genes, which are mutated in 30% of all cancers [106]. Mutations in the KRAS gene are frequently found in colorectal cancer, resulting in constitutive activation of the pathway [107]. Activating mutations in the KRAS gene are also found in a significant proportion of pancreatic ductal adenocarcinoma. Mutations in the EGFR are observed in non-small cell lung cancer (NSCLC) [108]. Due to the central role of the Raf/MEK/ERK signaling pathway in the initiation, maintenance, and metastasis of cancer, elements of this pathway (RAF, MEK, and ERK) are attractive targets for the development of potential cancer therapies. Some FDA-approved inhibitors against components of this pathway are used to treat several types of cancer, including melanoma, colorectal cancer, renal cell carcinoma, hepatocellular carcinoma, thyroid cancer, and glioblastoma [96,97,98,99,101,106]. BRAF inhibitors, such as vemurafenib and dabrafenib, have been approved for the treatment of BRAF-mutant melanoma, showing significant clinical benefits [109,110]. MEK inhibitors, such as trametinib, have been approved for the treatment of metastatic colorectal cancer with KRAS mutations in combination with other targeted therapies [107]. It is worth noting that the role of the RAF/MEK/ERK pathway in cancer can be complex, and its involvement may vary depending on the specific type and subtype of cancer, as well as the presence of other molecular alterations.

The aim of our study is to determine the role of the activation of PI3K/AKT/mTOR and RAF/MEK/ERK pathways in the pathogenesis of pituitary tumors. Despite numerous studies on prognostic parameters related to adenoma subtypes, the pathogenesis of pituitary tumors is still not fully understood, and the factors responsible for invasiveness, aggressiveness, and potential to metastasize are unknown. Moreover, there are no conclusive data on managing aggressive pituitary tumors that are resistant to standard treatment. Therefore, we try to summarize information on the use of novel therapies in the management of pituitary tumors, which, despite a few cases reported in the literature so far, seem to have significant value in the management of this disease. A more precise therapeutic approach will be possible by investigating and determining the expression of the receptors that are molecular targets for novel therapies, which may lead to improved outcomes for patients with pituitary tumors.

## 2. Materials and Methods

**Search strategy.** This study was performed according to the PRISMA (Preferred Reporting Items for Systematic Reviews and Meta-Analyses) guidelines for systematic reviews [111]. The search strategy included terms relevant to mTOR, the Raf/MEK/ERK pathway, and pituitary tumors and was conducted on three databases (PubMed, Embase, and Scopus) with a date filter of 2012–2023. The following search algorithm was used: (mTOR OR mTOR pathway OR PI3K/AKT/mTOR) AND (pituitary tumor OR pituitary adenoma) and (Raf/MEK/ERK pathway OR Raf OR MEK OR ERK) AND (pituitary tumor OR pituitary adenoma). Two independent researchers (AD and MS) performed the literature search. Figure 2 shows the flow of the study selection.

**Inclusion and exclusion criteria.** Original clinical studies published in English investigating the mTOR and Raf/MEK/ERK pathway signaling in tissue material or cell cultures were deemed eligible for inclusion. Exclusion criteria were (a) articles published in languages other than English; (b) narrative or systematic reviews and meta-analyses; (c) case reports, errata, comments, perspectives, letters to the editor, or editorials that did not provide any primary patient data; (d) published abstracts with no available full text; and (e) studies that included patients or tumors with unclear/undetermined histology. The publication date filter applied was 2012–2023.

**Data extraction.** The extraction of the following data was performed: the first author’s name, year of publication, the country in which the study was conducted, number of cases and controls, age, gender, method, expression of components of mTOR and Raf/MEK/ERK pathways, tumor-specific data, and cell culture-specific data.

## 3. Results

### 3.1. PI3K/AKT/mTOR Pathway in Pituitary Adenomas (Table 1)

The PI3K/AKT/mTOR pathway is involved in numerous vital cell functions, including cell-cycle regulation, growth of the cell, protein synthesis, and cellular metabolism [112]. The PI3K/AKT/mTOR pathway has been shown to be overexpressed in both hormonally active and inactive pituitary adenomas compared to a normal pituitary gland. Its expression is also described to be elevated in invasive pituitary tumors [113]. It has been shown that mTOR inhibitors are effective in pituitary adenomas. They reduce the proliferation and viability of the cells [114]. Comprehensive genomic profiling can reveal significant genomic alterations that have the potential of being targeted therapies for individual patients, including using of mTOR inhibitors [115]. Detailed data of PI3K/AKT/mTOR pathway expression in pituitary adenomas are shown in Table 1. The analyzed studies contained both expression in tissues as well as in cell cultures and the effect on the cells of an inhibiting mTOR signaling pathway. In a study on 53 pituitary samples obtained from patients (including GHomas, NFPAs, ACTHomas), mTOR kinase activity, estimated as pS6rp/eIF4E ratio, was elevated in pituitary adenomas and especially active in GHomas, which had the highest level of mTOR activity, which was statistically significant in comparison to NFPA [116]. However, there was no significant correlation found between the level of mTOR activity and any of the parameters: tumor volume, tumor largest dimension, Knosp’s grading, Ki-67%, and pErk activity [116].

Concerning prolactinomas, the most common functional pituitary adenomas, the treatment with dopamine agonists usually leads to normalization of the serum prolactin level and shrinking of the tumor mass. However, approximately 20% of patients do not respond to bromocriptine, and about 10% of patients to cabergoline. The main predictive factors for dopamine agonist resistance seem to be the male gender and the invasiveness of the tumor [117]. Bromocriptine induces apoptosis of prolactinoma cells through the ERK/EGR1 signaling pathway, and cabergoline leads to autophagy of prolactinoma cells by inhibiting the AKT/mTOR signaling pathway [118]. The study on cell culture has indicated that levels of p-AKT and p-mTOR were significantly lower in the group treated with cabergoline than in the control group [118]. The results of another research study on rat prolactinoma cell lines examined the effect of coadministration of ACT001 and cabergoline (CAB) [119]. The cell viability of the CAB + ACT001 group was significantly lower than the CAB and ACT001 groups themselves, and the number of autophagosomes in this group was higher than in the CAB group and ACT001 group [119]. The study showed that inhibiting the mTOR signaling pathway in GH3 cells induces cell death through autophagy. Adding this targeted therapy may help overcome the resistance to dopamine agonists [119].

Treatment of recurrent pituitary adenomas and infiltrating carcinomas is usually a clinical challenge due to resistance to conventional therapies. Moreover, involving surrounding structures often make it impossible to complete resection, and identifying new therapies that effectively restrict tumor growth is crucial. It was reported that everolimus (inhibitor of mTOR pathway) was effective in therapy for refractory ACTHoma with STK11 mutation resulting in stable disease for at least 6 months. [120] Another research study conducted on 95 pituitary adenomas analyzed the correlation between the gene transcript expression of mTOR; its protein complex facilitators, RAPTOR (regulatory associated protein of mTOR) and RICTO (rapamycin-independent companion of mTOR); and invasion, staging, and tumor growth of pituitary adenomas. A correlation between RICTOR expression and tumor size was found [113]. Moreover, mTOR expression was significantly associated with both RICTOR and RAPTOR, whose levels were significantly correlated [113]. It revealed a significant prognostic and predictive value of mTOR for patients. The research conducted on GH3 and GH4C1 rat pituitary adenoma cell lines showed that IGF1 improves somatotroph cell viability through the PI3K/AKT/mTOR pathway and the inhibitory effect of IGF1 on GH secretion via this pathway [121]. The data confirm that IGF-1 inhibits GH secretion at the transcriptional level through the PI3K/AKT/mTOR pathway. The results also indicate the effectiveness of mTOR inhibitors in reducing GH secretion by reducing somatotroph cell viability [121]. The study on 11 gonadotrophin adenomas obtained from patients showed broad genome analysis of lncRNAs and mRNA expression profiles using RNA-seq analysis [122]. Co-expression involving 126 lncRNAs interacting with 14 mRNAs of the mTOR pathway was found, which may support the pathogenesis of the gonadotrophin tumor [122]. The research examined the effect of everolimus, alone or with coadministration of cabergoline, on reducing cell proliferation in non-functioning PitNETs [123]. It showed that everolimus inhibited cell proliferation only in 5 out of 14 pituitary adenomas, so about two-thirds of NF-PitNETs in the study were resistant to this treatment; moreover, almost all tumors of this group were resistant to cabergoline [123]. However, coadministration of cabergoline caused a significant reduction in cell proliferation in 7 out of 9 tumors that were unresponsive to everolimus itself [123].

**Table 1 ijms-24-10952-t001:** PI3K/AKT/mTOR pathway in pituitary adenomas.

Study	Country	Clinical Data	Method and Sample	Results
Sajjad et al., 2013 [116]	Poland	53 Pituitary adenomas: 14 GHomas, 33 NFPAs, 6 ACTHomas Male/Female GHomas: 4/10 NFPAs: 21/12 ACTHomas: 2/4 Age: 54.6 ± 15.7	Tissue samples Western blot Primary cell culture	The level of mTOR kinase activity was calculated as pS6rp/elF4E ratio in all tissue samples. GHomas had the highest level of mTOR activity in comparison to NFPA (*p* = 0.04). The level of mTOR activity did not show any significant correlation with any of the parameters (tumor volume, tumor largest dimension, Knosp’s grading, Ki-67%, and pErk activity). All primary cell culture lines showed mTOR inhibition in response to rapamycin.
Di Pasquale et al., 2018 [121]	Italy	Not applicable	Cell culture (GH3 and GH4C1 rat pituitary adenoma cell lines) GH secretion RNA extraction and qRT-PCR Western blot	GH3 cell viability was significantly induced by IGF1 (+30%; *p* < 0.01 vs. untreated control cells) and reduced by everolimus and NVP-BEZ235 up to 30% (*p* < 0.01 vs. untreated control cells). GH4C1 cell viability was not influenced by IGF1 but was significantly reduced by everolimus (−60%; *p* < 0.01 vs. control treated cells) and by NVP-BEZ235 (−46%; *p* < 0.01 vs. untreated control cells). IGF1 significantly inhibited GH secretion (−40%; *p* < 0.01 vs. untreated control cells). IGF1 reduced GH mRNA expression (−37%; *p* < 0.01 vs. untreated control cells).
Zhu et al., 2021 [119]	China	Not applicable	Rat prolactinoma cell lines (GH3 cells) Western blot Cell counting kit (CCK)-8 assay	ACT001 induced autophagic cell death in cabergoline CAB-resistant GH3 cells by AMPK-mTOR pathway. The cell viability of the CAB + ACT001 group was lower than that of the CAB group (35.30 ± 3.33% vs. 59.63 ± 1.76%, *p* < 0.001) and ACT001 group (35.30 ± 3.33% vs. 84.10 ± 3.90%, *p* < 0.001). The number of autophagosomes in the CAB + ACT001 group was significantly higher than that in the CAB group (*p* < 0.001) and ACT001 group (*p* < 0.001).
Tang et al., 2019 [118]	China	Not applicable	Cell culture (MMQ cells and GH3 cells) Western blot	Levels of p-AKT (*p* = 0.0034) and p-mTOR (*p* = 0.0005) were significantly lower in the group treated by cabergoline than in the control group.
Mangili et al., 2022 [123]	Italy	NF-PitNETs:14 Male/Female: 6/8 Age: 60.7 ± 13.8	Tissue samples Primary cell culture RT-PCR analysis	Everolimus treatment was effective in reducing cell proliferation in 5 out of 14 NF-PitNET primary cultured cells (−39.2 ± 25.8% at 1 nM, *p* < 0.01 vs. basal). In NF-PitNETs resistant to Everolimus, the coadministration of cabergoline was effective in inhibiting cell proliferation in 7 out of 9 tumors (−31.4 ± 9.9%, *p* < 0.001 vs. basal).
Jia et al., 2013 [113]	China	95 Pituitary adenomas: NFPAs 59 PRLomas 5, GHomas 8, LHomas 2, FSHomas 2, TSHomas 4, ACTHomas 6, Mixed 9 Male/Female: 51/44 Age: <45 45 >45 50	Tissue samples Quantitative gene transcript analyses	Correlation between RICTOR expression and tumor size, namely *p* = 0.0012 and *p* = 0.0055 for tumors 1–2 cm and tumors >3 cm compared with tumors < 1 cm. Higher levels of mTOR were seen in tumors with cystic lesions (*p* = 0.044). Levels of mTOR were found to be significantly correlated with levels of both RAPTOR (*p* = 0.000234) and RICTOR (*p* = 0.0000002). RAPTOR and RICTOR levels were also significantly correlated (*p* < 0.0000002).
Li et al., 2017 [122]	China	11 Gonadotrophin adenomas Male/Female: 7/4 Age: 43.9 ± 16	Tissue samples RNA-seq analysis qRT-PCR	Genome-wide analysis of lncRNAs and mRNAs obtained from gonadotrophin adenomas. Co-expression involving 126 lncRNAs interacting with 14 mRNAs of the mTOR pathway (PCC > 0.80, *p* < 0.001), which might promote the pathogenesis of the gonadotrophin tumor.

Healthy controls (HC), real-time polymerase chain reaction (RT-PCR), quantitative polymerase chain reaction (qPCR), western blot (WB), immunohistochemistry (IHC), tissue microarray (TMA), growth hormone-producing pituitary adenoma (GHPA), ACTH-producing pituitary adenoma (ACTHoma), nonfunctional pituitary adenoma (NFPA), nonfunctional pituitary neuroendocrine tumor (NF-PitNET), LHoma (LH-producing pituitary adenoma), FSHoma (FSH-producing pituitary adenoma), TSHoma (TSH-producing pituitary adenoma), PRLoma (PRL-producing pituitary adenoma), phosphorylated-AKT (pAKT), long non-coding RNA (lcnRNA).

### 3.2. Raf/MEK/ERK Pathway in Pituitary Adenomas (Table 2)

Detailed data of Raf/MEK/ERK pathway expression in pituitary adenomas are presented in Table 2. In included studies, we analyzed both expressions in tissues and cell cultures. In research on pituitary samples obtained from patients with growth hormone-producing adenomas, the overexpression of EGFL7 positively correlated with the activation of EGFR (p-EGFR) [124]. The expression level of EGFL7 and p-EGFR in invasive GHPAs was much higher than in non-invasive GHPAs. Noticeably, EGFL7 knockdown significantly inhibited the activation of EGFR and downstream signaling pathways, including p-ERGR, p-AKT, and p-ERK. These data show that EGFL7 is a potential regulator of the EGFR pathway and plays an important role in migration and invasion of invasive GHPAs [124]. This study [125] examined the expression levels of EGFR and its downstream signaling molecules in fifty-two sporadic pituitary adenoma specimens and six normal pituitary glands. EGFR and its pathway signaling molecules had a higher expression in pituitary corticotroph adenomas than in normal pituitary glands. The EGFR levels significantly correlated with recurrence status and a short disease-free interval [125]. ERK was activated in most pituitary samples, including control samples [116,126]. In NFPAs, the activity of pErk showed a medium level of inverse correlation with Knosp’s grading [116]. However, the p-ERK expression was highly increased in prolactinomas compared with normal pituitary, demonstrating a strong activation of ERK1/2 signaling in prolactinomas [126]. In cell cultures, activated PI3K signaling strongly blunts ERK1/2-mediated PRL promoter activity, specifically via CDK4 [126]. ERK and PI3K have antagonistic actions in lactotrope cells, and it is functionally significant because Ras/ERK1/2-mediated PRL promoter activity is diminished during PI3K-driven cell-cycle progression. Inhibition of ERK1/2 signaling resulted in a small but significant increase in GH4T2 cell proliferation and colony formation [126]. Inhibition of PRL signaling increased ERK1/2 phosphorylation in a time-dependent manner. PRLR antagonist may increase ERK1/2 phosphorylation to decrease apoptosis, suggesting that the effect of PRL on apoptosis is mediated through ERK1/2 pathway inhibition [127]. USP8 mutations were shown to be associated with unbalanced EGFR signaling and enhanced phosphorylation of the downstream effector ERK1/2. Overexpression of USP8 G664R resulted in high levels of active phosphorylated ERK1/2 after 48 h of EGF stimulation, thus confirming that the novel USP8 variant sustains EGFR-MAPK signaling to promote corticotroph ACTH production and cell growth [119]. The duration of MAPK activation also appears to be critical in regulating cell differentiation vs. proliferation. This study [128] showed that Ras/MAPK-driven differentiation of pituitary precursor cells to a lactotrope phenotype may represent the mechanism for physiological lactotrope differentiation as well as a pathological mechanism for amplified lactotrope differentiation and secretion of prolactin in prolactinomas. Activation of Ras/MAPK signaling is not sufficient to drive lactotrope proliferation, but activation of pMAPK leads to GH4 pituitary somatolactotrope differentiation to a lactotrope phenotype. In pituitary cells, oncogenic mutations in Ras or Raf alone drive hyperplasia and delayed, benign adenoma formation, but are not sufficient for transformation [128].

**Table 2 ijms-24-10952-t002:** Raf/MEK/ERK pathway in pituitary adenomas.

Study	Country	Clinical Data	Method and Sample	Results
Sajjad et al., 2013 [116]	Poland	53 patients Male/Female: 27/26 Age: 54.6 ± 15.7 14 GHPA, 33 NFPAs, 5 ACTHomas	Tissue/cell culture IHC, WB	Erk was activated in most pituitary samples, including control samples Erk activity was the highest in control pituitary samples (*p* = 0.003) In NFPAs, the activity of pErk showed a medium level of inverse correlation with Knosp’s grading (*R* Spearman= −0.31, *p* = 0.018) In GHomas, pErk showed a strong level of correlation with somatostatin receptor subtype 2 A (SSTR2A) expression (R Spearman = 0.57, *p* = 0.04)
Liu et al., 2019 [125]	China	52 patients with CD (22 with tumor recurrence, 30 without) Age: 35.2 ± 12.4 6 HC	Tissue IHC, WB	EGFR immunoreactivity in 29 of 52 (55.8%) pituitary corticotroph adenomas (14 EGFR-positive adenomas in 20 (70%) recurrent adenomas and 15 EGFR-positive adenomas in 32 (46.9%) non-recurrent adenomas) and in 1 of 6 (16.6%) normal pituitary glands EGFR levels in the recurrent corticotroph adenomas were significantly increased compared to those in the non-recurrent ones p-EGFR and p-Erk were upregulated in recurrent adenomas but were not upregulated in non-recurrent adenomas or in normal pituitary glands, while the total Erk, total AKT, and p-AKT levels were unchanged EGFR protein was found to be significantly associated with the recurrence status (*p* = 0.005), cortisol level (*p* = 0.009), and ACTH level (*p* = 0.008) but was not related to the sex, age, or symptom duration of the patient (*p* = 0.280, *p* = 0.351 and *p* =0.142, respectively).
Liu et al., 2018 [124]	China	48 GHPA	Tissue/ Cell culture- Rat GHPA cell GH3 and mouse GT1.1 pituitary adenoma cells, WB, IHC, TMA	EGFL7 positive staining in invasive GHPAs was significantly higher (2-fold higher) than that in non-invasive GHPAs positive staining of total EGFR was higher in invasive GHPAs than that in non-invasive GHPA tissues average expression level of p-EGFR in invasive GHPAs was 3.5-fold higher than that in non-invasive GHPAs positive staining of p-EGFR was closely related with high-level EGFL7 in invasive GHPAs knockdown of EGFL7 expression significantly suppressed p-EGFR expression in GH3 cells p-AKT and p-ERK expression was decreased in EGFL7 knockdown cells after 48 h treatment with 50 ng mL−1 rhEGFL7 the level of EGFR, AKT, and ERK phosphorylation in GH3 cells was significantly increased, as compared with PBS control knockdown of EGFL7 effectively suppressed activation of EGFR signaling cascades in GH3 cells, including p-EGFR, p-AKT, and p-ERK
De Dios et al., 2019 [127]	Argentina	Not applicable	Cell culture—GH3 somatolactotrope cells, RT-PCR, WB	ERK1/2 inhibition mediates the apoptotic effect induced by PRLR activation in GH3 cells Inhibition of PRL signaling increased ERK1/2 phosphorylation in a time-dependent manner
Roof et al., 2018 [126]	USA	4 prolactinoma samples and 4 HC	Tissue/Cell culture—HEK 293T and BOSC cells GH4C1 rat somatolactotrope cells, WB, RT-PCR	p-ERK1/2 was undetectable in normal pituitary sample, in prolactinoma samples, p-ERK1/2 was expressed in all samples All prolactinoma samples and HC expressed t-ERK1/2 p-ERK/t-ERK expression ratio was increased in prolactinoma samples compared with normal pituitary tissue inhibition of the ERK1/2 signaling pathway promotes a decrease in the PRL/GH ratio inhibition of ERK1/2 signaling resulted in a small but significant increase in GH4T2 cell proliferation and colony formation ERK and PI3K signaling is dysregulated in human prolactinoma inhibition of Raf/MEK/ERK signaling increases GH4T2 cell proliferation
Booth et al., 2014 [128]	USA	Not applicable	Cell culture- GH4T2 cells	the duration of MAPK activation is critical in dictating the biological response activation of pMAPK leads to GH4 pituitary somatolactotrope differentiation to a lactotrope phenotype increase in the PRL to GH ratio observed both in vitro and in vivo suggests the differentiation of GH4 somatolactotrope cells into a lactotrope phenotype
Treppiedi et al., 2021 [129]	Italy	Not applicable	Cell culture—Murine pituitary corticotroph tumor cells, AtT-20 cells (ATCC CRL-1795™) WB	EGF is able to stimulate ERK phosphorylation in WT USP8 transfected cells was transient with a peak of phosphorylation reached at 24 h and strongly reduced at 48 h incubation in cells expressing S718del and G664R USP8, a persistent activation of ERK was observed at 48 h incubation with EGF (*p* < 0.05) Overexpression of USP8 G664R resulted in high levels of active phosphorylated ERK1/2 after 48 h of EGF stimulation, thus confirming that the novel USP8 variant sustains EGFR-MAPK signaling to promote corticotrophs ACTH production and cell growth

Healthy controls (HC), real-time polymerase chain reaction (RT-PCR), quantitative polymerase chain reaction (qPCR), western blot (WB), immunohistochemistry (IHC), tissue microarray (TMA), growth hormone-producing pituitary adenoma (GHPA), ACTH-producing pituitary adenoma (ACTHomas), nonfunctional pituitary adenoma (NFPA), phosphorylated-ERK (pERK), total-ERK (tERK) epidermal growth factor receptor (EGFR), epidermal growth factor (EGF), epidermal growth factor-like domain 7 (EGFL7), mitogen-activated protein kinase (MAPK), phosphorylated-mitogen-activated protein kinase (pMAPK).

### 3.3. Treatment

Most of the pituitary adenomas (PAs) demonstrate a favorable clinical course. Still, even in the case of non-functioning PAs, up to 50% of them regrow during the first 10 years after the initial operation [130]. In patients in whom observation was chosen over the surgical approach, tumor growth was noticed in 20% [131]. First-line treatment is usually surgery, excluding lactotroph tumors, which respond well to pharmacotherapy. However, a group of PAs are classified as aggressive based on their invasiveness, rapid tumor growth, resistance to treatment, and multiple recurrences despite standard approaches, including surgical, pharmacological, and radiotherapy treatment [3]. The incidence of aggressive pituitary tumors (APT) and carcinomas (PC) is hard to estimate, but it is suggested that around 1% of macroadenomas will exhibit aggressive behavior [131]. It can show up more than 10 years after diagnosing a pituitary tumor, and so far, there are no markers able to predict this. Current treatment options for APT and PC are insufficient to control tumor growth and often confer significant morbidity [10].

Therefore, it is vital to develop novel therapeutic strategies to address the unmet needs of patients with APT. Recently, several genomic studies have concentrated on the role of PI3K/AKT/mTOR and RAF/MEK/ERK pathways in pituitary carcinogenesis. Consequently, promising therapeutic agents have come to light. In this part of the article, we described current treatment options considering PI3K/AKT/mTOR and RAF/MEK/ERK pathways and highlighted novel compounds interfering with them.

#### 3.3.1. mTOR Inhibitors

Everolimus is an mTOR pathway inhibitor. There are no randomized control trials concerning the efficacy of this drug in APT. To date, only seven patients receiving everolimus have been described. Treatment response was noted only in two of them. The first one was APT-PRL treated with everolimus and cabergoline, where a partial response was achieved [132]. Stable disease for 5 months was observed in PC-ACTH treated with everolimus plus capecitabine. In those two patients, abnormalities in the mTOR signaling pathway were demonstrated. In the remaining cases, progressive disease was observed despite treatment [11,133,134,135,136].

#### 3.3.2. Tyrosine Kinase Inhibitors

Receptor tyrosine kinases (RTKs), which are situated on cell surface, bind molecules such as epidermal growth factor (EGF), fibroblast growth factor (FGF), and vascular endothelial growth factor (VEGF). It leads to tyrosine kinase (TK) phosphorylation and initiation of intracellular signaling through P13K/AKT/mTOR and Raf/MEK/ERK pathways. The result is a cell-cycle progression. Tyrosine kinase inhibitors (TKI) are orally administered drugs that reduce TK phosphorylation of target proteins, disrupting signaling pathways. A few studies show that functional regulation of pituitary tumor growth and hormonal secretion respond to TK inhibition, both in vivo and in vitro experiments [136]. Results from those clinical studies are summarized in Table 3.

#### 3.3.3. MAPK Inhibitors

ERK is a one of the mitogen-activated protein kinases (MAPK). Its activation leads to stimulation of cell proliferation and growth. Drugs targeting the Raf/Mek/ERK signaling pathway may constitute promising strategy in APT. Some of the currently used drugs for the treatment of PAs influence Raf/Mek/ERK pathways. Somatostatin analogs (SSTs) can exhibit their anti-tumor effect via inactivation of that pathway. While octreotide influences both ERK and PI3K/AKT, pasireotide acts possibly only on ERK [137]. In addition, dopamine inhibited proliferation of lactotroph pituitary cells via the ERK pathway [138]. Moreover, bromocriptine induced cell death by apoptosis via the ERK/EGR1 signaling pathway. On the other hand, cabergoline triggered autophagic death mediated by the AKT/mTOR pathway [118].

The mechanism of action of different inhibitors is shown in Figure 3. Interesting therapeutic approaches concerning Raf/Mek/ERK and PI3K/AKT/mTOR pathways are described in Table 4.
ijms-24-10952-t003_Table 3Table 3Interesting therapeutic approaches embracing the PI3K/AKT/mTOR pathway and the Raf/Mek/ERK pathway in aggressive pituitary tumors.Drug NamePublicationDrug MechanismMaterialResultscelastrol Cai et al.(2022) [139]second-generationmTOR inhibitorIN VITROACTH-secreting adenoma cell lines AtT20 IN VIVO-on mouse AtT20tumor xenograftsIN VITRO- blockade of cells in GO/G1 phase- induction of apoptosis and autophagy through downregulation of AKT/mTORIN VIVO-decrease in tumor volume and weight in micebuparlisibNVP-BEZ235Chanal et al. (2016) [140]dual PI3K/mTOR inhibitorIN VITRO-GH3 cell lines-Human prolactinomas in primary cell cultureIN VIVOOn rat SMtTW3 tumor xenograftIN VITRONVPBEZ235: -GH3 cell lines: induction of apoptosis, and cytostatic effect by accumulation of cells in G1-reduction in cell viabilityand hormone secretion in primary cell cultureBuparlisib: GH3 cell lines: limited effectprimary cell culture: limited effectIN VIVONVPBEZ235: -no effect on tumor growthBuparlisib:-decrease in tumor weightmetforminJin et al. (2018) [141]antihyperglycemic agent anti-tumor mechanism mainly includes the activation AMPK, thus inhibiting the mTOR pathway IN VITRO-AtT20 cell linesIN VITROInhibition of proliferation and induction of apoptosis by ac-activating AMPK/mTOR and inhibiting IGF-1R/AKT/mTOR NVP-BEZ235everolimusLee et al. (2011) [142] dual PI3K/mTOR inhibitormTOR inhibitorIN VITRO- GH3 cell lines- embryonic primary fibroblast cells-rat pituitaryadenoma cells in primary cultureIN VITRONVP-BEZ235 -Inhibition of PI3K pathway upstream and downstream of AKT- triggering of apoptosis due to decreasing AKT and S6 phosphorylation - reduction in cell viability more effective than everolimusACT001Zhu et al. (2021) [119].attenuation of the function of MnSODincrease ROS concentration in tumor cellsIN VITRO-GH3 and MMQ cell linesIN VITRO- Possible reversal of CAB resistance in GH3 cells by inhibition of mTOR signaling pathway and induction of cell death attributed to autophagy - possible reversal of BRC resistance in MMQ cells by activation of EGR1 signaling pathway and induction of celldeath due to apoptosis.XL765 with temozolomideDai et al. (2013) [143]dual-PI3K/mTOR inhibitorIN VITRO-GH3, T3-1, and MMQ cell linesIN VIVOOn rat GH3 tumor xenograftIN VITRO- synergistic inhibition ofgrowth of cell lines and induction of apoptosisIN VIVO-synergistic inhibition of tumor growthNelfinavir and radiationZeng et al. (2011) [144]Radiosensitizer HIV protease inhibitorIN VITRO-GH3, MMQ and AtT20 cell lines IN VIVOOn rat GH3 tumor xenograftIN VITRO-sensitization of PA cells to radiation, resulting in increased apoptosis-inhibition of the PI3K-AKT-mTOR pathway.IN VIVO-synergistic negative effect of radiotherapy and nelfinavir on tumor growthBIM-23A760Peverelli et al. (2010) [145]dopamine-somatostatin chimeric compoundIN VITROhuman non-functioning pituitary tumors cells in primary cultureIN VITRO-Activation of ERK1/2 and p38 pathways- antiproliferative and the pro-apoptotic effects on the cells fulvestrantGao eta al. (2017) [146]antiestrogenTissue samples289 PAs casesIN VITROGH3 and JT1-1 cell linesIN VIVOrat model of prolactinoma (injection of 17b-estradiol)Tissue samples-estrogen receptor alpha present in more than 50% of casesIN VITRO-Reduction in cell viability IN VIVO- inhibition of tumor growth by modulation of PTEN/MAPK signaling, including ERK pathwayMammalian target of rapamycin kinase (mTOR), phosphatidylinositol 3-kinase (PI3K), reactive oxygen species (ROS), manganese-superoxide dismutase (MnSOD), cabergoline (CAB), bromocriptine (BRC), early growth response protein 1 (EGFR-1), human immunodeficiency viruses (HIV), Phosphatase and tensin homolog (PTEN), mitogen-activated protein kinase (MAPK).


## 4. Discussion

### 4.1. Raf/MEK/ERK and PI3K/AKT/mTOR Pathways Are Involved in Pituitary Tumorigenesis and Aggressiveness

Raf/MEK/ERK and PI3K/AKT/mTOR cascades are key signaling pathways involved in pituitary tumorigenesis. They are associated with increased cell proliferation and survival in pituitary tumors. However, it is still unclear which pathway plays the most critical and central role in all cell and tissue types of pituitary adenomas. The data presented in this study [126] suggest that ERK and PI3K are both activated in prolactinoma and that there is a counterregulatory system between ERK and PI3K in lactotrope cells. This may indicate that both ERK and PI3K must be activated to promote lactotroph tumorigenesis or that ERK is activated in response to tumorigenesis. In GH3 cells, ERK1/2 inhibition mediated the apoptotic effect induced by PRLR activation [127]. The Raf/MEK/ERK signaling pathway is apparently related to cell growth and GH expression. A study in a rat somatotroph adenoma cell line demonstrated that GH3 cell proliferation was significantly increased after treatment with SDF1b, which induced phosphorylation of ERK1/2 [151]. MAPK, an intracellular mediator for CXCL12/CXCR4 in GH3 cell proliferation, seems to play an essential role in GH production and secretion. Interestingly, the duration of MAPK activation seems to be critical in dictating the biological response. Activation of pMAPK lead to GH4 pituitary somatolactotrope differentiation to a lactotroph phenotype [128]. In GH4C1 rat somatolactotrope cells, inhibition of Raf/MEK/ERK signaling increased cell proliferation [126]. Inhibition of ERK1/2 signaling resulted in a significant increase in GH4T2 cell proliferation and colony formation, while activation of dopamine D2 receptor (short isoform) with cabergoline stimulated ERK1/2 and resulted in reduced GH4C1 cell proliferation and ERK activation. It provides evidence that ERK activation might be a mechanism through which dopamine maintains lactotroph homeostasis [126]. In addition, the PI3K/AKT/mTOR pathway seems to play a crucial role in GHomas. In a study on 53 pituitary samples obtained from patients (including GHomas, NFPAs, ACTHomas), mTOR kinase activity, estimated as pS6rp/eIF4E ratio, was elevated in pituitary adenomas and mainly active in GHomas, which had the highest level of mTOR activity, which was statistically significant in comparison to NFPA [116]. PI3K/AKT/mTOR signaling is also involved in the effectiveness of pharmacological treatment. Analyzed studies on cell cultures indicated that levels of p-AKT and p-mTOR were significantly lower in the group of patients with prolactinoma treated by cabergoline than in the control group [118] and that inhibiting the mTOR signaling pathway in GH3 cells induced cell death through autophagy, and the addition of this target therapy may help to overcome the resistance to dopamine agonists [119].

Since BRAF gene mutations, which are a component of the RAF/MEK/ERK pathway, are commonly found in melanoma and papillary thyroid cancer, few studies have investigated its presence in PA [33,152,153,154]. In 50 human pituitary adenomas, including 25 NFPA and 25 secreting adenomas (10 GH, 5 PRL, 6 LH and/or FSH, 4 GH/PRL), only one V600E mutation in a NFPA sample was found, suggesting that B-RAF mutations are a rare event in pituitary tumorigenesis [153]. Another study investigated 37 PA for a mutation at the V600E position, and none was identified. Interestingly, B-Raf mRNA was overexpressed in pituitary adenomas compared to normal pituitary glands, especially in NFPAs. NFPAs also showed very variable expression of B-Raf protein [154]. That altered activity for the B-Raf/MAK/ERK pathway may play a pivotal role in the pathogenesis of these tumors.

### 4.2. The Activation of EGFR-Signaling Cascades Plays an Important Role in Cell Proliferation, Migration, and Invasion in PitNETs

Epidermal growth factor (EGF) and EGFR are expressed in functioning and non-functioning PitNETs, with higher expression in more aggressive tumor subtypes. The ErbB2 receptor is detected in all tumor subtypes, particularly in invasive tumors [155]. The existing preclinical and clinical evidence mainly concerns lactotrophic and corticotropic tumors [156]. The most common of the ErbB receptors is EGFR, expressed in 75% of human corticotroph tumors [157]. The effect of ErbB pathway signaling on lactotroph tumorigenesis was demonstrated in the pituitaries of female transgenic mice. Circulating PRL levels in *EGFR* and *HER2* transgenic mice were increased, and inhibiting *EGFR* or *HER2* signaling with oral lapatinib suppressed circulating PRL by 72% and attenuated tumor PRL expression by 80%, and also attenuated downstream tumor *EGFR/HER2* signaling [158]. It shows the role of ErbB receptors in prolactinoma tumorigenesis and the feasibility of targeting these receptors in treating refractory prolactinomas.

Patients with high levels of *EGFR* have increased tumor invasion and lower total resection than those with low levels of *EGFR* expression. A few studies show that functional regulation of tumoral growth and hormonal secretion respond to EGFR TKI inhibition, both in vivo and in vitro experiments [118,159]. *EGFR* expression is more common in the hormonally active pituitary adenomas than in the non-functioning ones [157].

Various studies showed that epidermal growth factor-like domain 7 expressions are highly elevated in multiple human cancers such as kidney tumors, malignant gliomas, hepatocellular carcinomas (HCCs), and colon, breast, and ovarian cancers [124,160,161,162]. EGFL7 is also overexpressed in GHPAs, and the expression is significantly correlated with pathologic characteristics, clinical progression, poor prognosis, and invasion [163,164]. The comparison of the expression level of EGFL7 in invasive GHPAs and non-invasive GHPAs demonstrated that EGFL7 expression in invasive GHPAs was much higher than that in non-invasive GHPAs, and overexpression of EGFL7 was positively correlated with activation of EGFR (p-EGFR) [124]. This evidence might suggest that EGFL7 is a potential regulator of the EGFR pathway and plays a vital role in the migration and invasion of invasive GHPAs.

### 4.3. Targeting Raf/MEK/ERK and mTOR Pathways as a Novel Therapeutic Approach

Targeting the RAF/MEK/ERK pathway has emerged as a potential therapeutic strategy for the treatment of pituitary tumors, particularly those that are hormone-secreting or refractory to conventional treatments. Cell-cycle regulation, in normal conditions, is controlled by complex intracellular signaling pathways, including the PI3K/AKT/mTOR and Raf/MEK/ERK pathways. In recent years, mTOR inhibitors have been approved for treatment of multiple cancers, including renal cell carcinoma, neuroendocrine tumors, and advanced breast cancer, and clinical trials are being conducted in other malignancies [165]. The upregulation and/or overactivation of the PI3K/AKT/mTOR pathway was reported in human PitNETs [166]. mTOR inhibitors showed anti-tumoral effects in in vitro human PitNET cultures as well as in in vitro and in vivo murine models and cell lines [140,167,168]. To date, the only inhibitor of the PI3K/AKT/mTOR pathway that has been studied in patients is everolimus (EVE), with seven reported cases so far [132,134,169]. In in vitro studies, EVE reduced non-functionating tumors’ cell viability by inducing apoptosis, with a mechanism likely involving IGF-I signaling but not VEGF secretion, suggesting that it might represent a possible medical treatment of invasive/recurrent non-functionating PitNETs [114]. Unfortunately, in vitro data demonstrating upregulation of PI3K/AKT/mTOR pathways in pituitary tumors have so far not translated into clinical success in aggressive pituitary tumors, apart from a single case report of a partial response to everolimus in an aggressive prolactinoma, where the patient achieved stability of tumor volume for 12 months and a decrease in prolactin levels [135].

### 4.4. Limitations and Further Perspectives

Several limitations of this systematic review need to be noted. Although we collected all published clinical evidence investigating mTOR and RAF/MEK/ERK pathway expression in pituitary tumors, the number of publications used for the systematic review was relatively small. The role of the RAF/MEK/ERK and PI3K/AKT/mTOR pathways in oncogenesis can be complex. Its involvement may vary depending on the specific environmental factors, type and subtype of pituitary tumor, and other molecular alterations. Pituitary tumors are a heterogeneous group; therefore, understanding the complex role of signaling pathways should be context-dependent. All studies included were published in English, and there might be publications in other languages that contain relevant results. The other restriction is limited information about the study group’s heterogenicity, including tissue material. Since the pathogenesis and predisposition to invasiveness is not fully understood, the topic requires further research with novel techniques to translate achieved results into clinical applications.

## 5. Conclusions

Taken together, although the Raf/MEK/ERK and PI3K/AKT/mTOR pathways play a pivotal role in the complex signaling network along with many interactions, further research is urgently needed to clarify the exact functions and the underlying mechanisms of these signaling pathways in the pathogenesis of pituitary adenomas and its role in invasiveness and aggressive clinical outcomes. Current treatment options for aggressive pituitary tumors are insufficient to control tumor growth and often confer significant morbidity. New data on cell signaling and molecules involved in cell proliferation and therapeutic targeting of molecular pathway shown to be involved in pituitary tumorigenesis may offer alternative treatments for these patients. Many questions in the context of the treatment of pituitary tumors remain unanswered. That is why further research in this area is urgently needed.

## Figures and Tables

**Figure 1 ijms-24-10952-f001:**
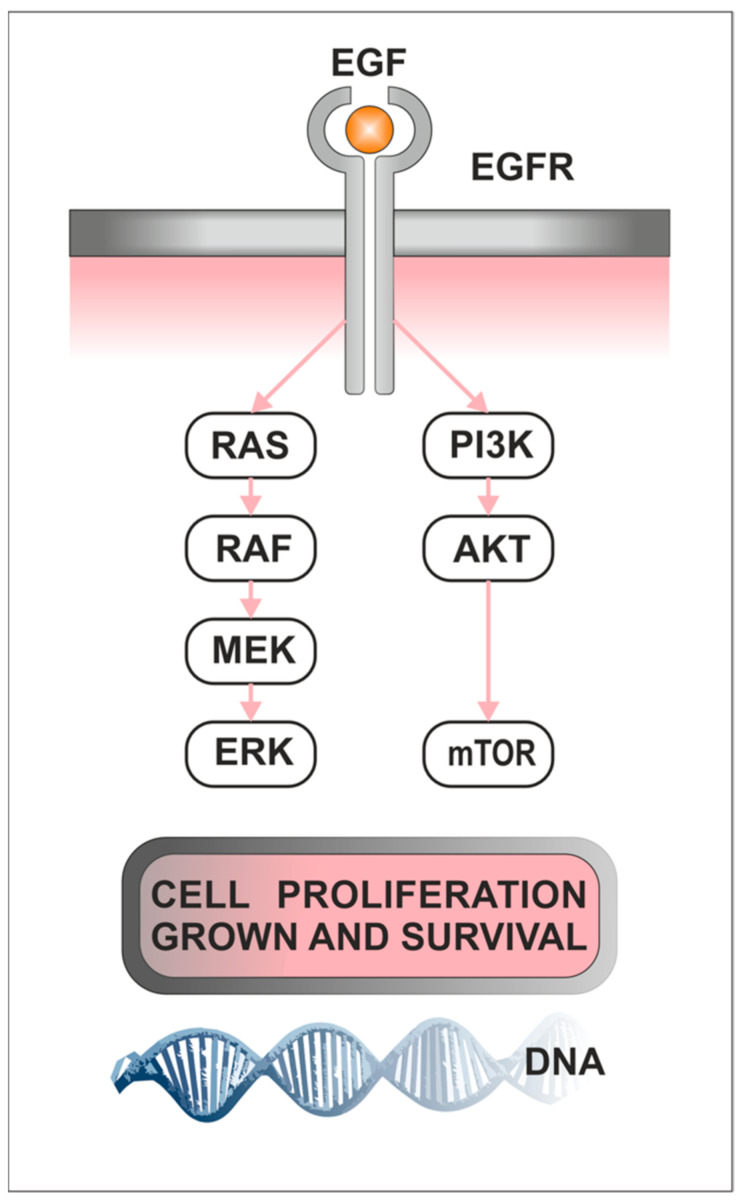
The biological role of the PI3K/AKT/mTOR and Raf/MEK/ERK signaling pathways in the development of cancer. Abbreviations: epidermal growth factor (EGF), epidermal growth factor receptor (EGFR).

**Figure 2 ijms-24-10952-f002:**
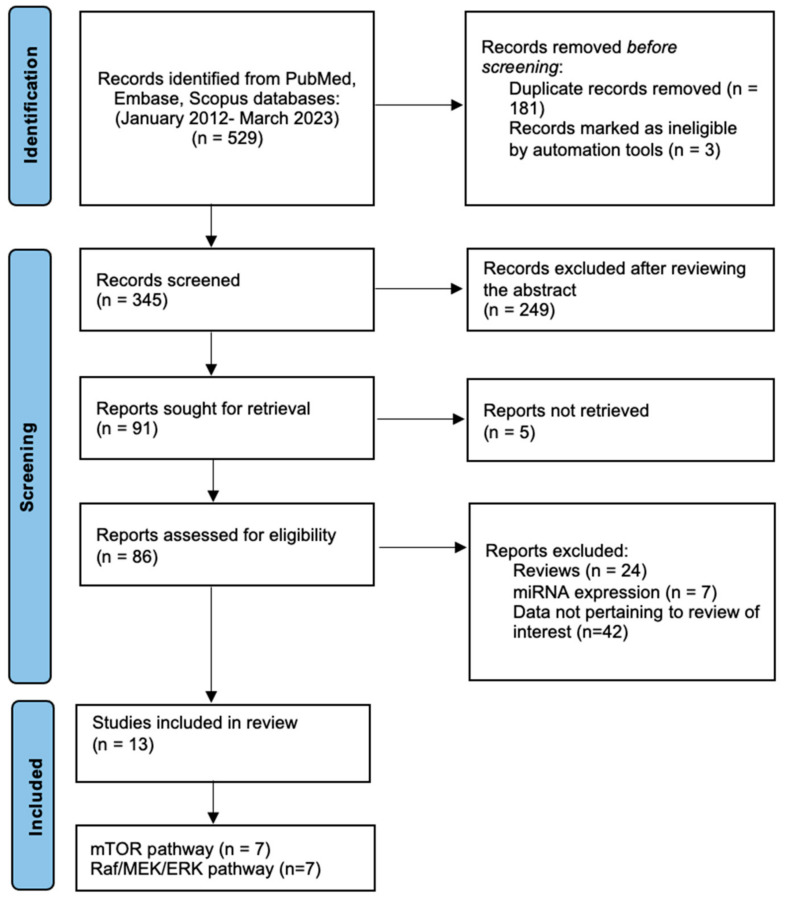
Flow diagram of the study selection.

**Figure 3 ijms-24-10952-f003:**
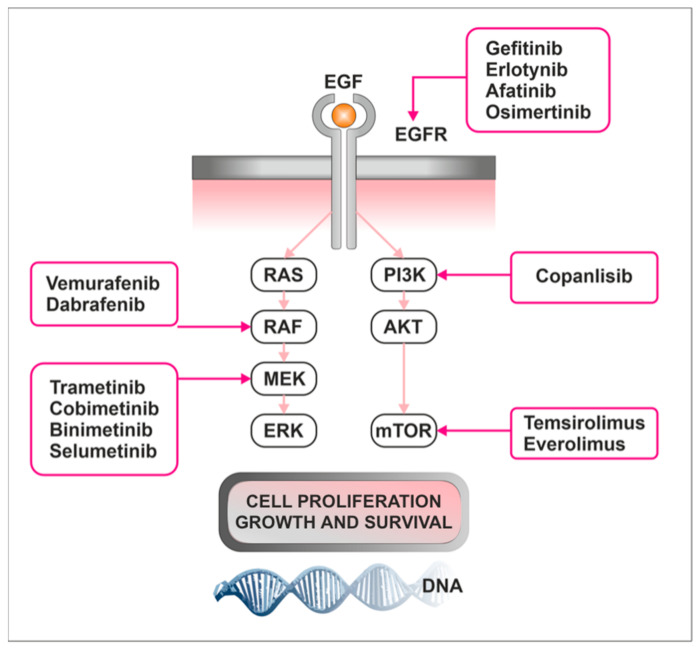
The Raf/MEK/ERK and PI3K/AKT/mTOR pathways and potential therapeutics. Different inhibitors target different mutations. They inhibit further signal transduction, ultimately preventing cell proliferation and survival or leading to apoptosis. Abbreviations: epidermal growth factor (EGF), epidermal growth factor receptor (EGFR).

**Table 4 ijms-24-10952-t004:** Growth factor-targeted drugs in aggressive pituitary tumors—clinical experience.

Drug Name	Publication	Drug Mechanism	Type of Tumor	Outcome
lapatinib	McCormack et al. (2018) [11], Cooper et al. (2019) [147], Cooper et al. (2021) [148],	dual EGFR and HER2/Neu inhibitor	10 APT-PRL; 2 APT unspecified	6 SD, 1 PR, 5 PD
erlotinib	McCormack et al. (2018) [11]	EGFR inhibitor	APT-ACTH	1 PD
gefitinib	McCormack et al. (2018) [11]	EGFR inhibitor	APT-PRL/GH	1 PR
sunitinib	McCormack et al. (2018) [11], Alshaikh et al. (2019) [135], Burman et al. (2022) [50]	oral multireceptor TK inhibitor	2 APT unspecified; 1 APT-ACTH	3 PD
apatinib plus temozolomide	Wang et al. (2019) [149]	VEGF inhibitor	APT-GH	1 CR
bevacizumab	Burman et al. (2023) [150]	recombinant monoclonal antibody blocking VEGF	8 APT and 4 PC	2 PR, 6 SD, 4 PD

EGFR—epidermal growth factor receptor; HER2/Neu—human epidermal growth factor receptor 2; VEGF—vascular endothelial growth factor; APT—aggressive pituitary tumor; PC—pituitary carcinoma, PRL-prolactin, ACTH—adrenocorticotropic hormone; GH—growth hormone; PR—partial remission; PD—progressive disease; SD—stable disease; CR—complete response; TK—tyrosine kinase.

## Data Availability

Not applicable.

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
