# Peer review of "The Role of Activation of PI3K/AKT/mTOR and RAF/MEK/ERK Pathways in Aggressive Pituitary Adenomas—New Potential Therapeutic Approach—A Systematic Review"

_ijms, 2023, doi:10.3390/ijms241310952_

Round 1
Author Response
Response to Reviewer 1 comments.
Dear Reviewer,
We appreciate you for your time in reviewing our paper. Attached is the revised manuscript with the requested changes as suggested by the reviewers. Below we provide point-by-point responses.
R1#1
The authors fail to lay a solid background of the disease PA and the PI3K/AKT/mTOR as well as RAF/MEK/ERK. What is the 5-year survival of PA? What is the most aggressive subtype? What is current treatment option for PA? PI3K/AKT/mTOR and RAF/MEK/ERK pathways are common in almost all kinds of cancer. What is the rationale of looking at these pathways in PA, or is there any thing specific about the involvement of these pathways in PA? Important logic step stones are missing here.
Response
We edited the introduction and added mentioned information (lines 39-40, 45-51, 67-83, 101-111).
R1#2
In section 1.1, the description about epigenetic modifications seems to be not suitable. The authors spend a whole paragraph describing what is epigenetic modifications instead of mentioning the core epigenetic modifications associated to PA. Please talk less about the generic information like DNA methylation, but spare more effort to talk about the details of the research.
Response
We edited the whole paragrath according to your instructions and mentioned core epigenetic modifications associated to PA (lines 113-167).
R1#3
Section 1.1, named as current knowledge of PA, is mainly about the epigenetic research. Please considering move the epigenetic information to a separate section or rename 1.1. Again, there is way more information could be included in under the term “current knowledge”. For example, the prevalence, treatment options, survival rate, prognosis etc.
Response
We edited the whole paragrath according to your instructions and mentioned core epigenetic modifications associated to PA (lines 113-167). The information about the prevalence, treatment options, survival rate, prognosis are provided in the introduction (lines 39-83). The detailed information about the novel treatment options are provided in section 3.3. (lines 565-640).
R1#4
The section 1.2 is not about “the challenge” at all. The authors should present the current challenges of treating this disease. For example, the drug response rate, the effectiveness of deriving preclinical models and animal models. And why some targets are hard to be targeted or not effective.
Response
We have highlighted current challenges concerning pituitary tumors. We discussed drug response rates and causes underlying insufficient drug efficacy. We described the challenges concerning preclinical and animal models of pituitary tumors (lines 169– 207).
R1#5
In 1.2, the authors also mentioned WNT and MAPK are important for PA. Again, it is intriguing why the authors set the boundary to be PI3k/AKT/mTOR and RAF/MEK/ERK but not looking at other equally important pathways. The current logic is not satisfying.
Response
We changed the whole paragraph 1.2 (lines 169- 207).
R1#6
Section 4.3 should be merged to Section 1.4, as they are not related to PA and thus pertain to introduction.
Response
Thank you for this comment. We merged the section according to your instructions (lines 325- 411).

Reviewer 2 Report
Abstract:
The abstract is well structured, concise, accompanied by 4 keywords that are well chosen and suggestive for the manuscript. This abstract is very clearly written and I think it opens this manuscript to a wider category of potential readers. I cannot find any fault with this abstract, so I give it maximum score.
On a scale of 1 to 10, I will give 10 points for the abstract.
Introduction:
The 6 pages of the introduction can be made into a stand-alone manuscript, in which the subject of this study is presented in detail, constituting a very good reminder or elaboration on this topic, as well as on the goal pursued in carrying out this systematic review. The bibliographic support of the text is particularly good. Definitely, grade 10 chapter.
On a scale of 1 to 10, I agree 10 points for introduction.
Methodology:
The methodology is correctly and completely presented, including in the form of a flow chart. Nothing to complain about, it deserves maximum score.
On a scale of 1 to 10, I must agree 10 points for methodology.
Results:
It seems that the manuscript continues at the same excellent level in the results chapter, which lives up to expectations, including through its graphic part (tables, figures). I see myself having to give the maximum score again...
On a scale of 1 to 10, considering the above statement, I agree 10 points for results.
Discussion:
The discussion chapter seemed a little softer, to put it this way. But, reading it carefully, I can't help to notice the extremely logical and concise approach to the 4 sub-points, as well as the honesty of the authors in sub-point “4.5. Limitations and further perspectives”. So...high score, again.
In this situation, on a scale of 1 to 10, I agree 10 points for discussion.
Conclusion:
I did not find anything to criticize in the conclusions chapter either.
On a scale of 1 to 10, I agree 10 points for conclusions.
Bibliography/References:
There are no less than 179 bibliographic references, current (considering the type of manuscript and the period of study), correctly written and correctly quoted in the text, which suggests an extremely detailed documentary on the chosen theme.
On a scale of 1 to 10, I agree 10 points for the bibliography.
Figures/Tables:
I identified 3 figures and 4 tables, of good quality, with satisfactory resolution, which are necessary and useful for the manuscript.
On a scale of 1 to 10, I agree 9 points for this chapter.
Review Decision:
Accept in present form.
Author Response
Dear Reviewer,
We appreciate you for your time in reviewing our paper. Thank you very much for your positive review and appreciation of our paper. Attached is the revised manuscript with the requested changes as suggested by the reviewers.
Round 2
Reviewer 1 Report
The manuscript has significantly improved.